# Hollow-tree super: A directional and scalable approach for feature importance in boosted tree models

Stephane Doyen⊙*, Hugh Taylor‡, Peter Nicholas‡, Lewis Crawford⊙‡, Isabella Young‡, Michael E. Sughrue⊙

Omniscient Neurotechnology, Sydney, Australia

⊙ These authors contributed equally to this work.
‡ These authors also contributed equally to this work.
* stephane.doyen@o8t.com

## Abstract

### Purpose

Current limitations in methodologies used throughout machine-learning to investigate feature importance in boosted tree modelling prevent the effective scaling to datasets with a large number of features, particularly when one is investigating both the magnitude and directionality of various features on the classification into a positive or negative class. This manuscript presents a novel methodology, "Hollow-tree Super" (HOTS), designed to resolve and visualize feature importance in boosted tree models involving a large number of features. Further, this methodology allows for accurate investigation of the directionality and magnitude various features have on classification and incorporates cross-validation to improve the accuracy and validity of the determined features of importance.

### Methods

Using the Iris dataset, we first highlight the characteristics of HOTS by comparing it to other commonly used techniques for feature importance, including Gini Importance, Partial Dependence Plots, and Permutation Importance, and explain how HOTS resolves the weaknesses present in these three strategies for investigating feature importance. We then demonstrate how HOTS can be utilized in high dimensional spaces such as neuroscientific setting, by taking 60 Schizophrenic subjects from the publicly available SchizConnect database and applying the method to determine which regions of the brain were most important for the positive and negative classification of schizophrenia as determined by the positive and negative syndrome scale (PANSS).

### Results

HOTS effectively replicated and supported the findings of feature importance for classification of the Iris dataset when compared to Gini importance, Partial Dependence Plots and Permutation importance, determining 'petal length' as the most important feature for positive and negative classification. When applied to the Schizconnect dataset, HOTS was able to

**Data Availability Statement:** All Iris files are available from the UCI Machine Learning Repository (https://archive.ics.uci.edu/ml/datasets/iris) All Schizconnect files are available upon

creating an account and loading the dataset from the Collaborative Informatics and Neuroimaging Suite (COINS) (https://coins.trendscenter.org/). After an account has been made with COINS, researchers can locate the Schizconnect data to download in its entirety by typing "COBRE" into the main search panel.

**Funding:** The authors SD & MS are co-founders of Omniscient Neurotechnology Pty Ltd. All authors (SD, HT, PN, LC, IY, MS) are employees of Omniscient Neurotechnology Pty Ltd. Omniscient Neurotechnology Pty Ltd provided support in the form of salaries for all authors (SD, HT, PN, LC, IY, MS). The study design, data collection and analysis, decision to publish, or preparation of the manuscript was decided collectively by the authors. The specific roles of these authors are articulated in the 'author contributions' section. The study is consistent with the objectives of Omniscient Neurotechnology Pty Ltd in furthering the scientific understanding of machine learning models in data analytics and neuroscience.

**Competing interests:** All authors (SD, HT, PN, LC, IY, MS) are employees of Omniscient Neurotechnology Pty Ltd. The study is consistent with the objectives of Omniscient Neurotechnology Pty Ltd in furthering the scientific understanding of machine learning models in data analytics and neuroscience. Related innovative elements of the methodology outlined in the Study are the basis for patent protection sought by Omniscient Neurotechnology Pty Ltd. This does not alter our adherence to PLOS ONE policies on sharing data and materials expressly discussed in the Study.

resolve from 379 independent features, the top 10 most important features for classification, as well as their directionality for classification and magnitude compared to other features. Cross-validation supported that these same 10 features were consistently used in the decision-making process across multiple trees, and these features were localised primarily to the occipital and parietal cortices, commonly disturbed brain regions in those afflicted with Schizophrenia.

## Conclusion

HOTS effectively overcomes previous challenges of identifying feature importance at scale, and can be utilized across a swathe of disciplines. As computational power and data quantity continues to expand, it is imperative that a methodology is developed that is able to handle the demands of working with large datasets that contain a large number of features. This approach represents a unique way to investigate both the directionality and magnitude of feature importance when working at scale within a boosted tree model that can be easily visualized within commonly used software.

## 1. Introduction

Tree based models, a category of supervised machine learning algorithms, have become widely used to perform regression or classification. Among the reasons for their popularity is the ability to perform predictions on data with high dimensionality, mixed type variables and complex, non-linear relationships—better so than linear methods [1].

In many real-life applications however, model interpretation is equally as valuable as the prediction output. Yet understanding why a prediction was made can be a non-trivial exercise given that tree-based models can become extremely complex (e.g. deep trees) and difficult to interpret at scale. Interpretability becomes even more complex in the case of boosted trees such as XGBoost [2] where numerous different trees are bagged together and weighted with different importance (boosting).

Some methods exist to understand why a model makes these predictions, such as Gini importance, partial dependence plots and permutation analysis [3–5]. Whilst those techniques provide a certain degree of useful insights into the model, they each lack one or more important properties for explainability of the model, specifically: (1) the direction in the relationship between features and the response variable (e.g., whether feature X1 is predictive of the negative/positive outcome) and (2) magnitude (e.g., how much feature X1 influences the prediction towards the positive or negative outcome). It is important to note that a successful technique would need to achieve (1) and (2) in a way that would scale to a larger number of features, thus providing a truly commensurable understanding of datasets, particularly those that have high dimensionality.

Recently, a method was proposed to linearize tree-based model nodes to provide an answer to both (1) and (2) [6]. However, this method has some limitations when it comes to applying it on boosted trees as each instance of the model added to the ensemble can have a different tree structure.

In this paper, we first explore the current predominant methods in the field which exist to determine feature importance in decision trees, before addressing a common problem within these methodologies: the ability to jointly derive magnitude and directionality of classification

between features in these models. We then present a novel feature contribution method, "Hollow-Tree Super" which extends the linearization methods available for single and ensemble trees to include boosted trees. Finally, we demonstrate and provide an example for how this new method can be used to analyse high-dimensionality data, such as in the case of human neuroimaging datasets investigating brain pathology.

## 1.1 Common approaches to feature importance for single trees

Before demonstrating our feature contribution method on boosted trees, it is important to first discuss the most common methods currently available for calculating feature importance. Whilst there exist many feature importance algorithms, here we examine three popular methods in the field: Gini importance, partial dependence plots and permutation importance. We do this on single decision trees—the simplest of the tree-based approaches and thus the most readily interpretable. For this (and throughout the worked example), we use the well-known iris dataset [7]—a simple classification problem with four input features relating to plant dimensions: petal length, petal width, sepal length and sepal width.

Typically, the iris dataset categorizes plants as one of; *iris versicolor*, *iris virginica* and *iris setosa*. To keep things simple, we removed the iris setosa group to make this a binary classification problem (*iris versicolor* being the negative class (= 0) and *iris virginica* as the positive class (= 1).

Using the sklearn DecisionTreeClassifier package [8] we constructed a single decision tree with a max depth of 4 (Fig 1a).

**1.1.1 Gini importance.** A standard approach to determining feature importance is to score features based on the number of times or probability a variable is utilized by the model for splitting, weighted by some other value. This could be the criterion used to select split points (Gini or entropy), or some other metric such as the squared improvement to the model's F-score. Fig 1b shows the feature importance's for our binarized decision tree, computed using sklearn's "feature_importances_" property. This function calculates feature importance using the "normalized" total reduction of the criterion brought by that feature, also referred to as the "Gini importance".

This approach attributes a score to each feature, where a higher value indicates greater influence on the output prediction. From our simple decision tree, the most important feature for outcome prediction was petal length, with petal width, sepal length and sepal width showing no significant difference in their influence for predicting outcomes through this tree.

Despite receiving scrutiny for biasing against variables with higher numbers of categories [9] the use of Gini importance has seen a resurgence in recent years, particularly when analyzing genomic datasets [10,11]. Whilst it remains an informative metric for assessing the relative importance of different features used in a model by providing them a relative ranking and is scalable, it lacks the ability to provide directional information on classification between features. Indeed, nothing is said for the variable "petal length" to be more predictive of the positive or negative outcome.

**1.1.2 Partial dependence plots.** As an alternative, partial dependence plots (PDP) are often used to visualize decision boundaries. They help to describe relationships with non-linear effects, and show interactions between features. To construct such a plot, the PDP function is calculated at each possible value of a feature, representing the average model prediction output at that value. Without the requirement for linearity, PDPs have shown great advantages in ecological studies, where features often interact non-linearly to influence species classification [12]. PDP functions can be calculated for one or two features [13], such that one can visualise how the values for one feature influence classification, or how values between two distinct

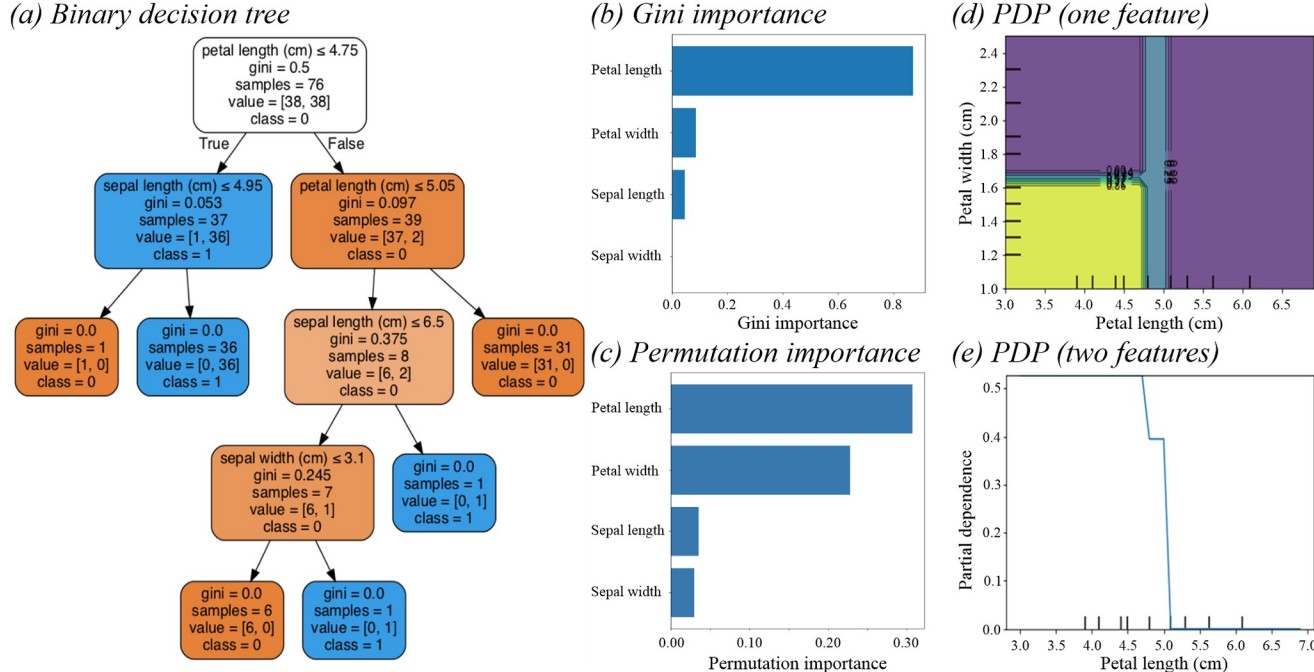

**Fig 1. Single binarized decision tree created in SKlearn to classify the Iris dataset. (a)** A single binarized decision tree was created in the sklearn DecisionTreeClassifier package to effectively delineate the Iris dataset. The depth four tree used Iris versicolor as the negative class (= 0), and Iris Virginica as the positive class (= 1). Iris Setosa was removed from the classification system to allow the binarization of the dataset. **(b)** We calculated the Gini Importance for our decision tree using sklearn's "feature_importances_" property. This revealed 'petal length' to be the most important feature in the model for classification into the positive or negative class. **(c)** When performing permutation importance on our four features included in our simple decision tree, we found that permuting the values of petal width had the greatest impact on the model prediction error when attempting to classify data to the positive or negative class. Unlike Gini importance, this analysis revealed that petal length was also significantly important in the classification process, whilst sepal length and width again were found to be relatively unimportant in the decision making process, and permutation of these features did not greatly impact model prediction error. **(d)** A one feature partial dependence plot (PDP) for 'petal length' revealed that positive classification as Iris Virginica (partial dependence) was non-linearly related to 'petal length', with a critical point at 5cm marking certain negative (Iris Versicolor) classification. **(e)** By introducing 'petal width' and conducting a two feature PDP, we were able to determine that these two features were increasingly dependent for positive classification at values lower than 5cm and 1.6cm for petal length and width respectively, giving us directional and magnitudinal inferences between two features (For two feature PDP's, a colour map is added to help visualise dependencies such that green indicates a greater partial dependency than purple).

features interact together to influence classification. Fig 1d shows the PDP for 'petal length' alone, and Fig 1e demonstrates the interaction between 'petal length' and 'petal width' and the predicted outcome of the constructed simple decision tree.

These two PDP's together reveal that petal length and petal width become dependent features for classification at values less than 5cm for petal length and 1.6cm for petal width respectively, and at greater values for either feature they are largely independent for classification. The benefits of PDP's are that they are easy to implement, interpret and provide a measure of directionality for feature importance. This would scale nicely for the iris dataset, which has four features. However, the main disadvantage is that the 2D representation of PDPs limits observations to two variables at a time [13]. This makes PDPs difficult to interpret at the scale of a dataset which comprises hundreds of variables, as each feature or pair of features at most, would require their own plot and subsequent analysis to ultimately determine and inform feature importance.

**1.1.3 Permutation importance.** Finally, permutation importance of features can be used to measure the change in the model's prediction error as the value of the feature is 'permuted'. Permutation is the process of shuffling the data points for one feature whilst retaining the

order for all other features to measure which variable most greatly affects model prediction error [14]. From this perspective, a feature is unimportant if changing its values has little effect on the model's error—implying that the model did not rely strongly on this feature to make predictions. Conversely, an 'important' feature would increase the model error when its values are shuffled.

We can use sklearn's permutation importance package to perform this calculation on the Iris dataset (Fig 1c).

Whilst this is another useful approach to determine the magnitude of feature importance, much like Gini importance, this method lacks information about directionality. In applying permutation importance analysis to the Iris dataset, we are able to glean that permuting 'petal width' had the largest effect on the model's prediction error, and was therefore the most 'important' feature. However, we are again unable to say whether 'petal width' was more predictive of the positive or negative outcome.

## 1.2 Direction and magnitude in feature importance coefficients

As shown, none of the three methods already described are able to provide information about both outcome directionality and magnitude, in a way which could be efficiently scaled to a large number of features.

Linearizing decision trees [6] offers the advantage of giving feature importance coefficients that have a direction, which magnitude can be compared and further scaled to larger datasets. A general linear equation can be derived from the model by considering that each decision in the tree stems from a feature and these decisions either increase or decrease the value from the parent node. Thus, it is possible to consider the final prediction as the sum of each feature's contribution within the tree plus a bias value (typically the topmost sample average).

To achieve this for a given prediction, the decision tree that led to that prediction is navigated and the local increments of feature contributions at each node (positive or negative) are identified. In this way, each prediction can be mathematically described using Eq (1).

$$f(x) = bias \sum\nolimits_{k=1}^{K} contribution\,(x, k) \tag{1}$$

Where; $K$ is the number of features, *bias* is the value at the root of the node and *contribution(x, k)* is the contribution from the k-th feature in the feature vector x [15].

This approach is similar to linear regression in a dynamic sense, and so by borrowing from regression models in this way we can achieve a similar level of interpretability as linear models. More specifically, this view of trees enables us to isolate the contributions of each feature for each prediction. It should be noted though that this linearization technique is limited in use to boosted trees whose input feature vectors only contain linearizable variables [16].

Here we provide a worked example, showing how the feature contributions are calculated for one prediction. Fig 2 depicts the decision tree, and Table 1 outlines the values of each feature for a sample iris plant, as well as the feature contribution score. Such score can easily be calculated programmatically though the Python package eli5 package [17] provides a convenient implementation which Fig 2 was derived form.

If we have a flower with the attribute values described in Table 1 (sepal length = 6.9, sepal width = 3.1 and petal length = 4.9), the model estimates the likelihood of this being in the positive class (y = 1, an iris virginica) at 1.0 (i.e. 100%).

Since this is a simple single decision tree, we can easily follow the path through the tree (Fig 2) for this prediction and isolate the relative contribution of each feature;

$$Contribution_{petal\;length\;(cm)} = (0.051 - 0.493) + (0.25 - 0.051) = -0.243 \tag{2}$$

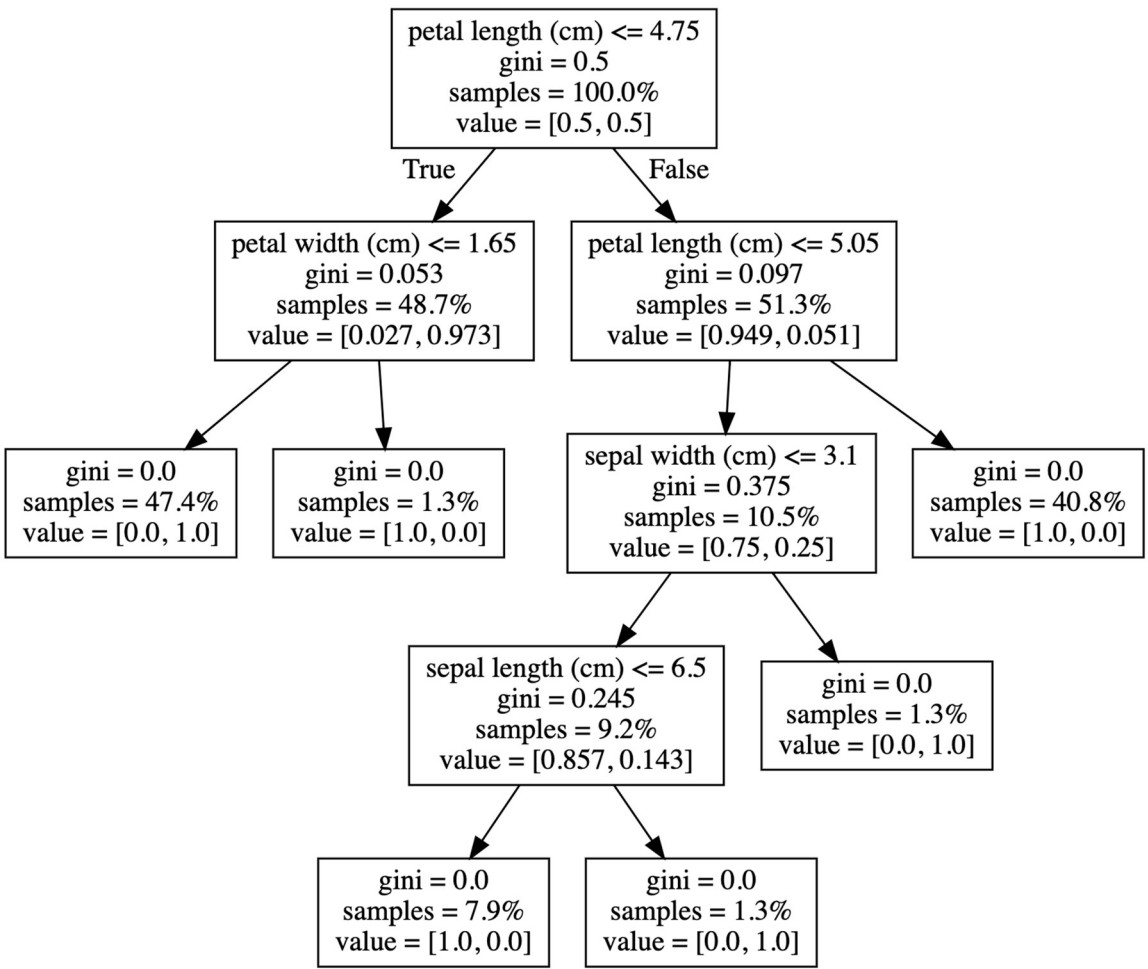

**Fig 2. Single decision tree linearized using the eli5 package.** By employing a general linear equation to define the relative contributions of each feature as decisions are made at an increasing depth in the tree, it is possible to derive feature importance values which lead to a positive (Iris Virginica) or negative (Iris versicolor) classification. This analysis offers a unique advantage over Gini importance, partial dependence, and permutation importance such that feature importance coefficients provide both a directionality and magnitude for each feature in delineating data into the positive and negative classes.

$$Contribution_{sepal\ width\ (cm)} = (0.143 - 0.25) = -0.107 \tag{3}$$

$$Contribution_{sepal\ length\ (cm)} = (1.0 - 0.143) = 0.857 \tag{4}$$

**Table 1. Sample prediction and feature contribution score.**

| Class = 1, probability = 1.0 | | |
| --- | --- | --- |
| **Contribution** | **Feature** | **Value** |
| +0.857 | Sepal length (cm) | 6.90 |
| +0.493 | BIAS | 1.00 |
| -0.107 | Sepal width (cm) | 3.10 |
| -0.243 | Petal length (cm) | 4.90 |

The model is telling us that sepal length of 6.9cm makes it much more likely that this is a iris virginica (85.7% more likely, to be exact), whilst the petal length of 4.9 somewhat lowers the chances of this being a iris virginica. This has made our single decision tree fully interpretable, in the sense that we can readily say exactly how each feature has influenced the prediction.

The final prediction (probability of being in the positive class) is the sum of the feature contributions towards the prediction (0.857–0.107–0.243 = 0.507) plus the bias value (= 0.493), in this case, 1.

Since we don't want to study feature importance at a prediction level, we can average the contributions of each feature across all predictions to obtain a global picture of importance. The method to achieve this involves three steps:

1. Filter out incorrect and low probability predictions.

2. Separate the feature contributions towards the positive and negative classes.

3. Sum the contributions of each feature across all predictions and normalize these values by dividing by the number of predictions made.

After performing this process on the Iris dataset, we found the top feature for predicting both the positive and negative classes to be petal length (Fig 2). Unsurprisingly, petal length also had the greatest Gini importance and the second greatest permutation importance, suggesting this method correctly captures feature importance within a decision model and supports the findings of alternative, less robust, analyses.

### 1.3 Scaling feature extraction from single tree to boosted ensembles

Ensemble trees and particularly boosted ensemble trees often provide a superior prediction ability than single trees [12]. Unfortunately, the linearization method described above falls short as each tree of the boosted ensemble leads to a different succession of split and group mean for each node due to the random start for the generation of each tree used in the ensemble. As a matter of fact, running the prediction function provided in the eli5 package would lead to a different linear equation for each tree of the ensemble.

To circumvent this issue, we propose here an aggregation method across several boosted tree instances within a model to provide directional, proportional, and interpretable feature importance. In addition, this method has the advantage of working across multiple cycles of cross-validation dealing more positively with boosted trees' tendency to overfit.

## 2. Methods

### 2.1 Gradient-boosting decision trees using XGB classifier

We use XGBClassifier [2] to fit the model. Table 2 below shows the improvement in model accuracy, ROC AUC and F1 score achieved by the Gradient boosted approach over single Decision trees. Accuracy is the proportion of correct predictions to total predictions. Accuracy can be represented in the following equation: Where TP is true positive predictions, TN is true

**Table 2. [Iris dataset] model performance metrics.**

| Measure | Decision Tree | XGBoost |
|---|---|---|
| Accuracy | 0.920 | 0.940 ± 0.075 |
| ROC AUC | 0.923 | 0.940 ± 0.075 |
| F1 Score | 0.917 | 0.939 ± 0.081 |

negative predictions, FP is false positive predictions, and FN is false negative predictions. "Predictions" refers to the number of items for which a classification was made by XGBClassifier, for example here, the number of plants in the iris dataset.

$$Accuracy = \frac{TP + TN}{TP + TN + FP + FN} \tag{5}$$

F1 score (or F score) is the weighted average of precision and recall and takes into account both false positives and false negatives. Precision is a measure of the number of predictions of the positive class that actually belong to the positive class. It is calculated as the number of true positives as a proportion of the sum of true positives and false positives. Recall is a measure of the number of positive class predictions out of all positive classes. It is calculated as the number of true positives as a proportion of the sum of true positives and false negatives.

ROC AUC provides an aggregate measure of performance across all possible classification thresholds. It is measured as the area under the receiver operating characteristic curve, which is a plot of the true positive rate (i.e. recall) to the false positive rate.

## 2.2 Performing Hollow-Tree Super (HOTS)

1. For each subject, we can use eli5 explain_prediction [17] to obtain each feature's contribution to each prediction (ignoring the bias value). Note that when extracting feature contribution from a boosted tree model, the 'weights' become the log odds contribution of each feature (as opposed to the probabilities shown under a single tree model).

2. We separate the weights contributing towards the positive and negative class cases.

3. Incorrect predictions, and those with a prediction probability of less than 70% are filtered out, keeping only the subjects that were correctly predicted by the model with confidence.

4. The weights across all the remaining predictions are aggregated by feature and divided by the number of predictions, obtaining an average weight of each feature per prediction.

5. As mentioned in (1), the weights provided are the log odds of being in the class that is ultimately predicted (i.e. for positive class predictions, the weights are the log odds of being in the positive class). Additionally, the weights are the log odds at the value of the feature currently being predicted. Thus, to extract the directionality desired here, it is necessary to infer the sign of these log odds for each feature. This is achieved by identifying whether the mean value for each feature in this positive class is greater or less than the mean of each

**Table 3. 5-fold feature contribution average for positive and negative classification of the Iris dataset using HOTS.**

| Feature | Classification | Fold 1 | Fold 2 | Fold 3 | Fold 4 | Fold 5 |
|---|---|---|---|---|---|---|
| **Petal Length** | Positive | -4.04 | -1.54 | -1.90 | -2.63 | -2.51 |
| | Negative | +2.97 | +1.69 | +1.24 | +2.90 | +2.60 |
| **Petal Width** | Positive | -1.09 | -3.86 | -2.31 | -2.42 | -2.26 |
| | Negative | +1.61 | +2.83 | +2.64 | +2.11 | +2.20 |
| **Sepal Length** | Positive | -0.13 | -0.13 | -0.11 | +0.14 | -0.04 |
| | Negative | +0.27 | +0.63 | +0.25 | -0.25 | +0.15 |
| **Sepal Width** | Positive | +0.02 | +0.13 | +0.01 | +0.02 | -0.25 |
| | Negative | -0.12 | -0.24 | +0.07 | -0.27 | -0.08 |

Note that a greater positive value indicates a greater contribution towards positive (*Iris Virginica*) or negative (*Iris Versicolor*) classification.

corresponding feature in the negative class. The log odds of each feature in the positive class are multiplied by the sign of the mean value for the positive class less the mean value for the negative class, whilst the log odds of each feature in the negative class are multiplied by the inverse sign of this same calculation. The 'weights' now become the log odds with a standard view of directionality. However, it is important to note that this approach assumes linearity between the feature values.

6. As mentioned, one challenge to this approach is the inherent instability of feature importance over different runs caused by the overfitting and random factor dependence of gradient boosted tree models. Different cuts of data for training and testing produce different results. This problem can be solved by performing cross-validation to arrive at a stable set of features. This allows for greater confidence when interpreting the model as it highlights only features that are consistent across runs. Here we use 5-fold cross-validation to obtain the mean feature importance's, with an average accuracy across all 5 folds = 0.94.

## 3. Results

### 3.1 Iris dataset

Using HOTS, A high degree of concordance with the features determined by Gini and permutation importance is maintained, with petal length and petal width both identified as the top two most predictive features (Fig 3). Since these weights are effectively the log odds for the respective classes, we can interpret them as such. The large negative value for petal length in the positive class (= -0.45) indicates that as petal length increases, the odds of being in the positive class decreases. A similar conclusion can be made for petal width. Further, we can see that the sepal dimensions had little to no predictive power.

We are also able to perform a count of how many folds each feature appears during the cross-validation process (Fig 4a). We see that all four features of the iris dataset were used to make predictions in each of the 5 folds (Fig 4b).

This is particularly useful when modelling on data with a large number of features, as it highlights the features that may only appear sporadically (low fold count in Fig 4b), but which have a large weight in those instances (large average feature importance weighting in Fig 4a).

This methodology unveils a way to achieve a similar outcome as calculating feature importance by Gini importance or improvement to F-score, whilst also making use of ensemble methods like gradient boosting which provide a superior fit. Further, it does this in a way that better quantifies the impact of the variable since the weights in Fig 4a represent the averaged feature importance contributions after cross-validation towards the respective final predictions.

This method is optimized for approximating the contribution of each feature to the classification outcome and while it is limited to one observation at a time, it can be easily scaled.

### 3.2 Case study: HOTS feature importance in Schizophrenic brain data

Furthermore, we propose this method as a suitable way of tying clinically observed behaviours to functional brain regions, or "parcellations".

A "parcellation atlas" in simple terms is a map of the brain. It delineates regions of the neocortex that exhibit similar properties across individuals, such as functional activity, structural connectivity, or cellular composition. Thus, a 'parcellation' is a region of the brain that expresses similar properties in a population, even if the exact boundaries or topological location may differ between individuals.

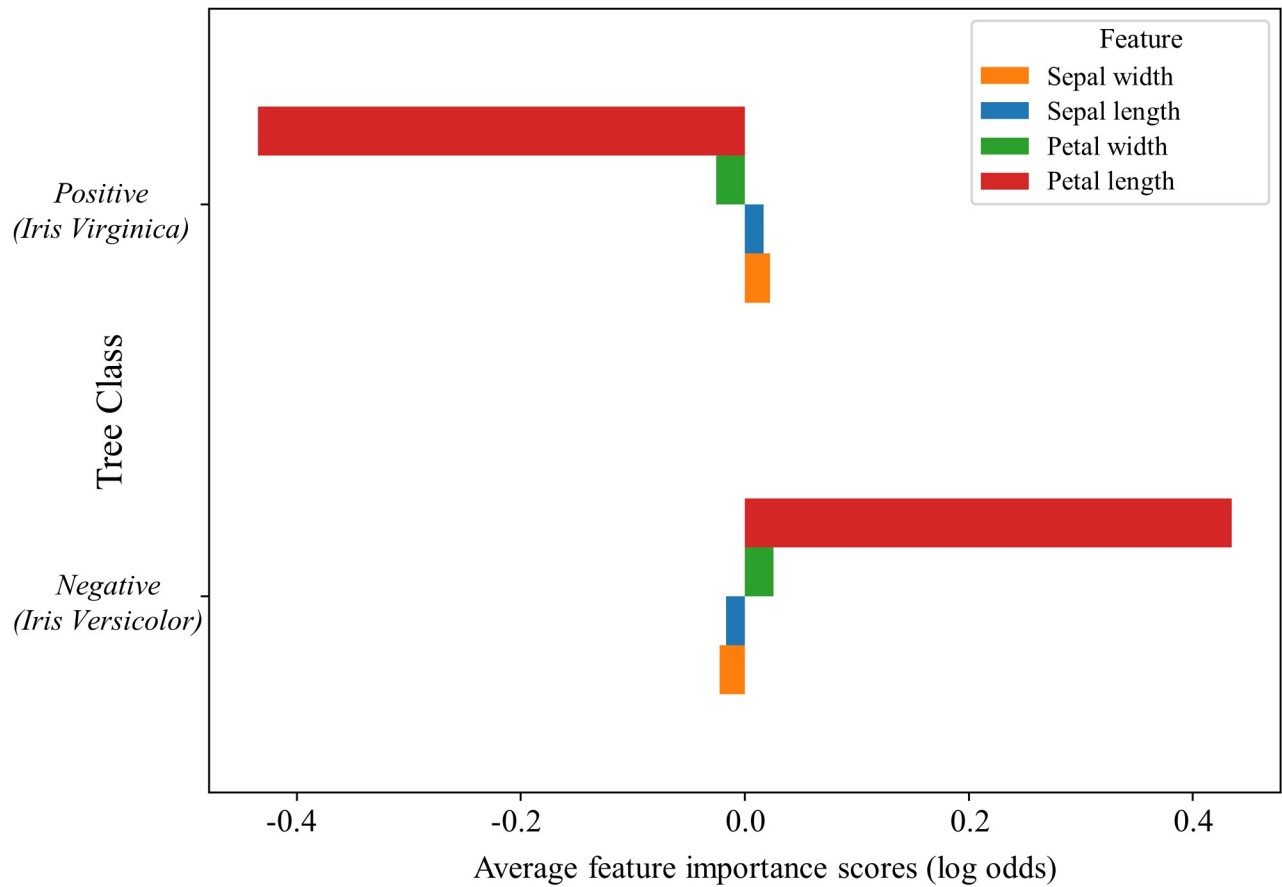

**Fig 3. Average feature contribution weight per prediction in a single decision tree.** After entering the Iris dataset into our linearized decision tree, we found the most important feature for successfully determining positive or negative class to be 'petal length'. Importantly, the outputs provided in this analysis are given a magnitude and direction for their respective involvement in classification compared to other features. These metrics offer a significant improvement on previous analyses, whilst remaining consistent with the findings of Gini and permutation importance, where 'petal length' had the highest and second highest weightings respectively. Note that this represents the output data from a single decision tree (fold) prior to cross-validation.

Parcellation atlases are particularly useful when analyzing functional magnetic resonance imaging (fMRI) data, which through recording changes in blood flow over time, can produce a representation of neural activity. This information can further be used to show functional connectivity between regions of the brain if the activity recorded between them exhibits a statistical relationship. A parcellation atlas can be used to reduce this complexity of pairwise correlations by reducing the comparisons to a finite number of regions, assumed to perform somewhat uniform functions.

Recently, Doyen et al. (forthcoming) developed a machine learning-based technique for parcellating the brain in a way that is both subject specific, and comparable between subjects [2,18]. Using this technique, we are able to generate adjacency matrices representing the correlation between every pair of parcellations—described for the remainder of this paper as 'connectomic features'.

The applied atlas contains 379 parcellations in Glassian nomenclature [19] equating to 71,631 input features, and so any method used to tie clinically observed behaviour to these

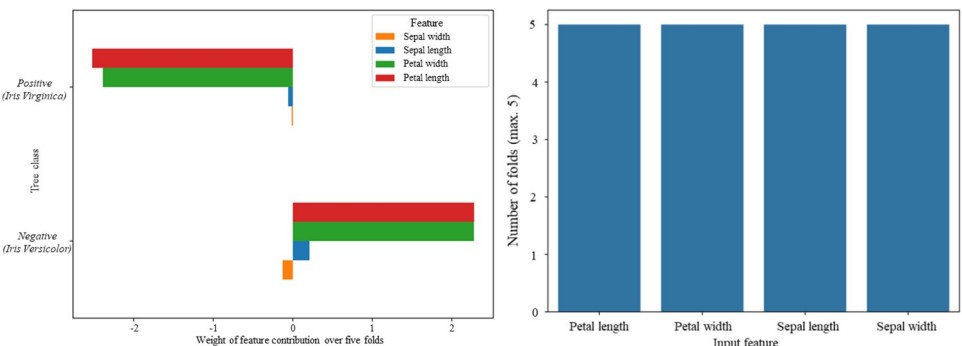

**Fig 4. Average feature contributions and count number over 5-fold cross validation. (a)** By investigating the five folds used for cross-validation shown in Table 3, we once again determined 'petal width' and 'length' to be the most important features for positive and negative classification within our decision tree. **(b)** A count of the number of folds each feature appeared in during the cross-validation process revealed that all four features appeared equally throughout the decision making process—suggesting that despite being significantly less important features for positive and negative classification than 'petal length' and 'width', sepal features are utilised equally regularly to make predictions within the model across multiple folds.

functional brain areas would not only need to provide direction and magnitude, but also be scalable to a large number of features.

Here we present a case example of the feature contribution for boosted trees method described throughout this manuscript, demonstrating its ability to effectively tie functional regions of the brain to clinically observed behaviours.

We performed this analysis using the SchizConnect Center for Biomedical Research Excellence (COBRE) dataset [20]. The dataset consists of the necessary brain MRI data for mapping, as well as neuropsychological assessment scores for 60 patients diagnosed with Schizophrenia. The analysis was designed as a binary classification problem—predicting the presence or absence of a specific symptom. Specifically, we use item N4 from the patient's Positive and Negative Syndrome Scale (PANSS) [21], which measures the degree to which patient's show "passive/apathetic social withdrawal". Each patient recorded their response on a 7 point Likert scale, representing increasing levels of psychopathology (1 = absent, 7 = extreme), with a sensible binarization point determined at a score of 2 (i.e. subjects with a N4 score > 2 were in the positive class, and ≤ 2 were in the negative class). As described above, the input features used to predict the presence or absence of this symptom were the subject's pairwise functional correlation between the 379 regions of the brain atlas which together form a large adjacency matrix.

As with the iris dataset, we use XGBClassifier [2] to fit the model and perform 5-fold cross-validation with an average accuracy of 0.71. Using the same feature contribution method described for the iris dataset, we are able to generate a list of the features (parcellations) that were most predictive of the positive and negative class symptom (Fig 6). The positive and negative class importance's are complementary, and so only the positive class (target) importance's are shown here. As Fig 5 shows, the region of the brain most predictive of the presence of 'passive/apathetic social withdrawal' is area 2 of the right parieto occipital sulcus (R_POS2). Since these values are represented as log odds, the value for R_POS2 (-1.33) tells us that as the values for this parcellation increased, the probability of being in the positive class (having greater "passive/apathetic social withdrawal") decreased. Conversely, we can see that increases in the value for the dorsal aspect of the left lateral intraparietal lobule (L_LIPd) indicated an increased probability of being in the positive class (+0.48).

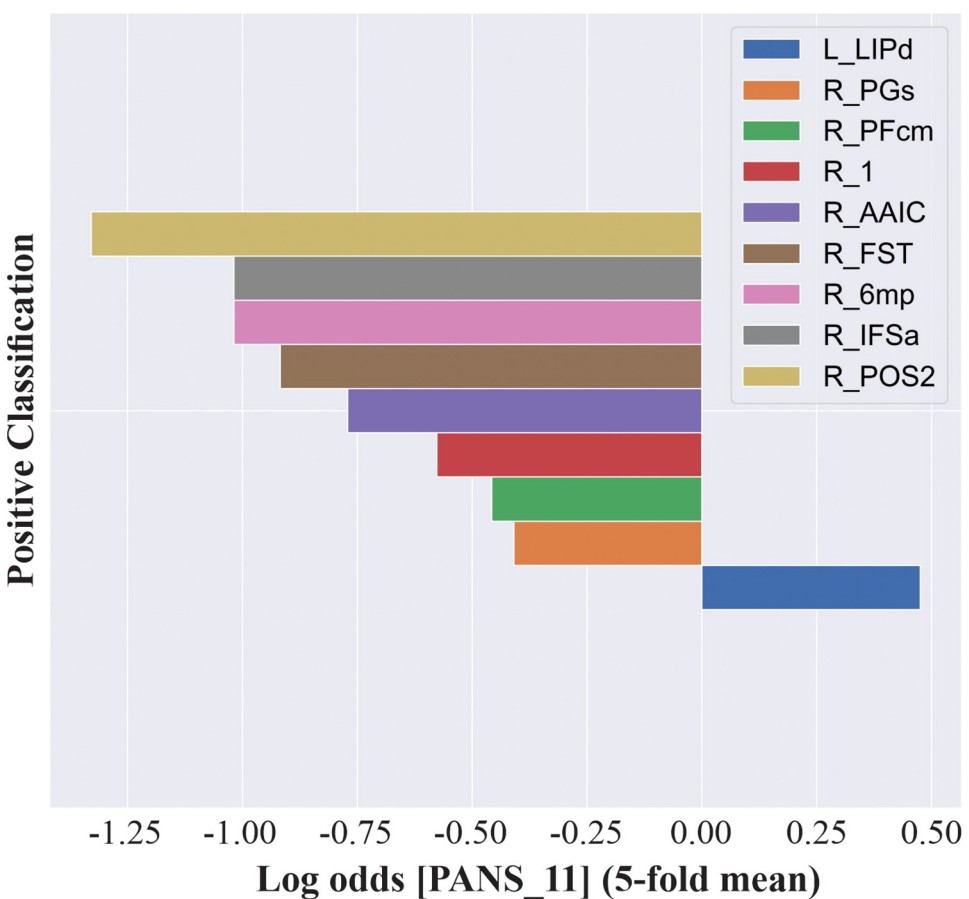

**Fig 5. Predictive brain regions for passive/apathetic social withdrawal.** Applying our methodology to a cohort of 60 subjects from the Schizconnect COBRE dataset revealed brain regions most predictive for item N4 of the Positive and Negative Syndrome Scale (PANSS). Connectivity matrices were generated between 379 cortical and subcortical parcellations using a scheme derived from Glasser and colleagues (2015), and feature importance was carried out on connectivity measures extracted from each individual parcellation. After performing cross-validation and averaging the weights of feature importance, we determined R_POS2, R_IFSa, R_6mp, R_FST, R_AAIC, R_1, R_PFcm, R_PGs, and L_LIPd to be most predictive for positive classification of PANSS N4. Positive class indicates a score > 2 on item N4 of the PANSS. X-axis values are provided in log odds to more easily visualise the features of importance on a logarithmic scale. L_ = left side, R_ = right side, POS2 = area 2 of the parietal-occipital sulcus, IFSa = anterior inferior frontal sulcus, 6mp = medial posterior aspect of area 6, FST = lateral occipital visual area, AAIC = anterior agranular insular cortex, 1 = primary sensory area, PFcm = centromedian part of parietal area F, PGs = superior aspect of parietal area G, LIPd = dorsal aspect of the lateral intraparietal area.

Further, cross-validation revealed that the R_POS2 was used by the boosted tree model to make predictions in 3 out of 5 folds (Fig 6). This suggests that this feature was consistently important for making predictions towards the positive class.

## 4. Discussion

Throughout this manuscript we detailed and validated a novel method, "Hollow-tree super" (HOTS), which accurately provides metrics of directionality and magnitude between features in boosted tree models. Further, the ability to discern between a large number of features and be scaled to large datasets makes HOTS an easily implementable method which can be utilized by a range of data-driven disciplines. When applied to the Iris dataset, HOTS showed high

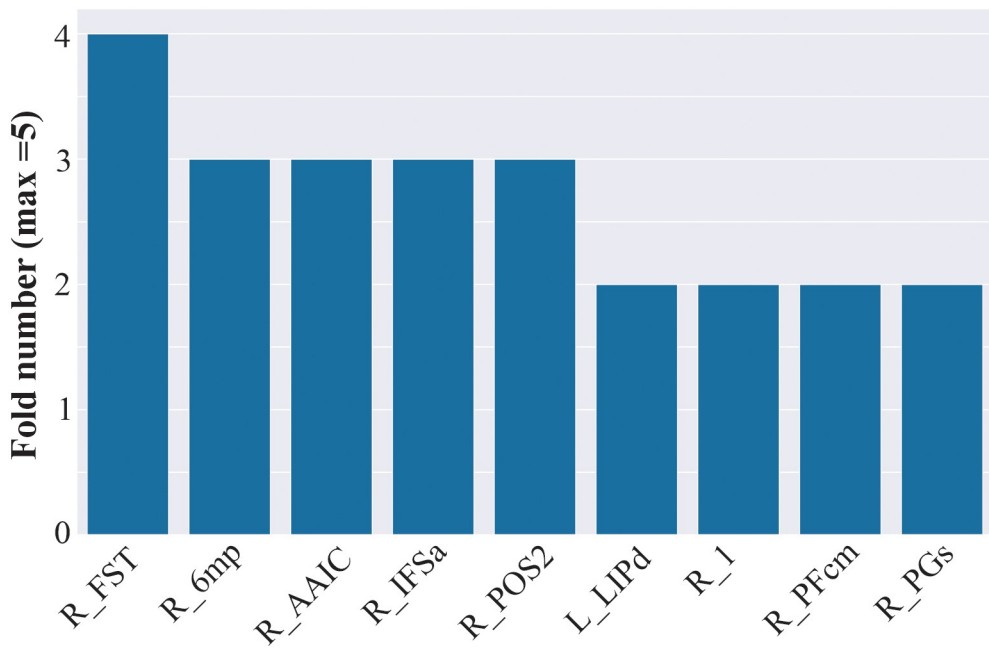

**Fig 6. Count of recurring features across folds.** By performing a count of feature appearance during the cross validation process, we determined that the same features (parcellations) responsible for positive class classification were also consistently used in 2 (R_PGs, R_PFcm, R_1, L_LIPd), 3 (R_POS2, R_IFSa, R_AAIC, R_6mp), or 4 (R_FST) out of the 5 folds, suggesting that these same features were regularly used throughout the decision making process. Note that this plot was abbreviated to only show features with a count of greater than 1.

concordance with other commonly used methods for investigating feature importance: Gini importance, partial dependence plots and permutation importance. In a neuroscientific setting, HOTS was able to accurately identify between 379 features, the most responsible brain regions for positive classification of Schizophrenia from item N4 on the PANSS. Cross-validation supported that these same features were consistently utilized throughout the decision-making process, improving our confidence that the chosen model had an accurate fit for diagnosis. Modern neuroscientific investigations further bolster our findings, supporting that abnormal activity and connectivity between occipital and parietal brain regions occurs during Schizophrenia [22,23]. That is, our use of HOTS supports the current understanding of the determinants of this pathology and provides neuroscientists with a novel method for investigating similar pathologies of the brain.

## 5. Study limitations

Importantly, this study involves a few key limitations that require addressing. First, our investigation involved benchmarking HOTS as a feature importance method by testing its concordance with Gini Importance, Partial Dependence Plots and Permutation Importance, as we felt those were the three most common methods used throughout the field. Whilst HOTS was able to effectively reproduce the results of these three methods using the Iris dataset, we cannot with certainty assert that HOTS would reproduce the results of other methodologies such as Information Gain or Chi-squared tests for feature selection. Additionally, as with any machine-learning model, increasing the size of the dataset would improve the ability of HOTS

to make accurate inferences about feature contribution and classification into positive and negative classes [24], and whilst the Iris dataset remains widely-accepted a valid tool for testing machine-learning algorithms [25,26], our observations on the Schizophrenic brain data may be improved with greater volumes of imaging data which we aim to collect and test in future investigations. Finally, an inherent limitation to boosted-tree modelling extends to our method in that instability can arise due to random factor dependence occurring between trees. Whilst we did consider this problem, we believe the incorporation of cross-validation when performing HOTS provides an effective regulating step to best minimize this, and further promotes this method as superior for investigating feature importance in boosted-tree models.

## 6. Conclusion and future directions

These results support that HOTS offers superior metrics for investigating feature importance than previously used methodologies. At least in the case of boosted-tree models, we suggest that HOTS could be well incorporated into the modelling process and provide thorough and easily interpretable metrics for the most responsible features which are also consistently utilized for classification. Overall, the potential to visualise directionality and magnitude of feature contribution within a single methodology streamlines the process of performing boosted-tree modelling, and offers a unique approach which can be applied to a range of different types of data, including high dimensional datasets such as those encountered in the study of neuro-imaging data. Further investigations should test the applicability of this method on other datasets, for example in neuroscience, where HOTS could be further utilized to investigate which brain regions are most responsible for other disorders of the brain such as Major Depressive Disorder, Alzheimer's and Parkinson's disease. A longitudinal dataset of pre- and post-treatment scans with corresponding symptom scores could be used to verify the results of these neuroscientific applications. An interesting future extension of the method could include designing an approach to deliver HOTS feature importance scores in regression models, as well as a method for working with categorical input data.

## Author Contributions

**Conceptualization:** Stephane Doyen, Isabella Young, Michael E. Sughrue.

**Data curation:** Stephane Doyen, Hugh Taylor, Peter Nicholas.

**Formal analysis:** Stephane Doyen, Hugh Taylor, Peter Nicholas.

**Investigation:** Stephane Doyen, Hugh Taylor, Peter Nicholas, Isabella Young.

**Methodology:** Stephane Doyen, Hugh Taylor, Peter Nicholas, Lewis Crawford, Isabella Young.

**Resources:** Isabella Young, Michael E. Sughrue.

**Software:** Stephane Doyen, Peter Nicholas.

**Supervision:** Stephane Doyen, Michael E. Sughrue.

**Validation:** Stephane Doyen, Michael E. Sughrue.

**Visualization:** Stephane Doyen, Lewis Crawford.

**Writing – original draft:** Stephane Doyen, Lewis Crawford.

**Writing – review & editing:** Stephane Doyen, Hugh Taylor, Lewis Crawford.

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
