## [Decision Letter · Decision Letter 0]

31 May 2021

PONE-D-21-06049

Hollow-tree Super: a directional and scalable approach for feature importance in boosted tree models

PLOS ONE

Dear Dr. Doyen,

Thank you for submitting your manuscript to PLOS ONE. After careful consideration, we feel that it has merit but does not fully meet PLOS ONE’s publication criteria as it currently stands. Therefore, we invite you to submit a revised version of the manuscript that addresses the points raised during the review process.

Based on the comments received from the reviewers and my own observation, I recommend major revisions for the paper.

We look forward to receiving your revised manuscript.

Kind regards,

Thippa Reddy Gadekallu

Academic Editor

PLOS ONE

Journal Requirements:

[The author(s) received no specific funding for this work ].   

We note that one or more of the authors are employed by a commercial company: Omniscient Neurotechnology

Reviewers' comments:

Reviewer's Responses to Questions

**Comments to the Author**

1. Is the manuscript technically sound, and do the data support the conclusions?

Reviewer #1: Yes

Reviewer #2: Yes

2. Has the statistical analysis been performed appropriately and rigorously? 

Reviewer #1: Yes

Reviewer #2: Yes

3. Have the authors made all data underlying the findings in their manuscript fully available?

Reviewer #1: Yes

Reviewer #2: Yes

4. Is the manuscript presented in an intelligible fashion and written in standard English?

Reviewer #1: Yes

Reviewer #2: Yes

5. Review Comments to the Author

Reviewer #1: 1. Kindly include a paragraph at the end of section 1 describing the flow and division of sections in the research paper.

2. The authors can add the formula for accuracy as an equation and explain the meaning of TP, TN, FP and FN in their problem.

3. The authors have done few literature works, but they haven’t compared any of them in their experiment. I recommend the authors should compare their work with at least 2 works present in the literature.

4. The authors should add few explanations why they have to choose GINI as there are lots of other feature selection algorithms such as IG, Chi, Pearson, and so on.

5. The authors can include a result analysis for each of the components (Gini Importance, Partial Dependence Plots, Permutation Importance)

Please cite the following papers

1. Ashokkumar P, Siva Shankar G, Gautam Srivastava, Praveen Kumar Reddy Maddikunta, and Thippa Reddy Gadekallu. 2021. A Two-stage Text Feature Selection Algorithm for Improving Text Classification. ACM Trans. Asian Low-Resour. Lang. Inf. Process. 20, 3, Article 49 (April 2021), 19 pages. DOI:https://doi.org/10.1145/3425781

2. Maddikunta, P. K. R., Gadekallu, T. R., Al-Ahmari, A., & Abidi, M. H. (2020). Location based business recommendation using spatial demand. *Sustainability*, *12*(10), 4124.

Reviewer #2: 1. The paper is written in a good manner. Some minor touches can improve this paper more.

2. The quality of the figures can be improved more.

3. The contributions of the authors are not clear. They have mentioned in first contribution.

4. Future research directions section is core, however, it is not good at all.

5. What are the computational resources reported in the state of the art for the same purpose?

- Please cite each equation and clearly explain its terms.

- Clearly highlight the terms used in the algorithm and explain them in the text.

6. What are the evaluations used for the verification of results?

7. Several paragraphs contain trivial information and should be dropped.

8. I found some English mistakes please check them.

9. Kindly refer the below paper:

1. Rajput, D.S., Basha, S.M., Xin, Q. et al. Providing diagnosis on diabetes using cloud computing environment to the people living in rural areas of India. J Ambient Intell Human Comput (2021). https://doi.org/10.1007/s12652-021-03154-4

6. PLOS authors have the option to publish the peer review history of their article (what does this mean?). If published, this will include your full peer review and any attached files.

Reviewer #1: No

Reviewer #2: No

---

## [Author Response · Author response to Decision Letter 0]

12 Jul 2021

Please see revised cover letter for updated funding and competing interest statement.

---

## [Decision Letter · Decision Letter 1]

4 Oct 2021

Hollow-tree Super: a directional and scalable approach for feature importance in boosted tree models

PONE-D-21-06049R1

Dear Dr. Doyen,

We’re pleased to inform you that your manuscript has been judged scientifically suitable for publication and will be formally accepted for publication once it meets all outstanding technical requirements.

Kind regards,

Sriparna Saha, PhD

Academic Editor

PLOS ONE

Additional Editor Comments (optional):

Reviewers' comments:

Reviewer's Responses to Questions

**Comments to the Author**

1. If the authors have adequately addressed your comments raised in a previous round of review and you feel that this manuscript is now acceptable for publication, you may indicate that here to bypass the “Comments to the Author” section, enter your conflict of interest statement in the “Confidential to Editor” section, and submit your "Accept" recommendation.

Reviewer #2: All comments have been addressed

2. Is the manuscript technically sound, and do the data support the conclusions?

Reviewer #2: Yes

3. Has the statistical analysis been performed appropriately and rigorously? 

Reviewer #2: Yes

4. Have the authors made all data underlying the findings in their manuscript fully available?

Reviewer #2: Yes

5. Is the manuscript presented in an intelligible fashion and written in standard English?

Reviewer #2: Yes

6. Review Comments to the Author

Reviewer #2: 1. The study presents the results of original research.

2. Results reported have not been published elsewhere.

3. Experiments, statistics, and other analyses are performed to a high technical standard and are described in sufficient detail.

4. Conclusions are presented in an appropriate fashion and are supported by the data.

5. The article is presented in an intelligible fashion and is written in standard English.

6. The research meets all applicable standards for the ethics of experimentation and research integrity.

7. The article adheres to appropriate reporting guidelines and community standards for data availability.

7. PLOS authors have the option to publish the peer review history of their article (what does this mean?). If published, this will include your full peer review and any attached files.

Reviewer #2: No

---

## [Editor Report · Acceptance letter]

15 Oct 2021

PONE-D-21-06049R1 

Hollow-tree Super: a directional and scalable approach for feature importance in boosted tree models 

Dear Dr. Doyen:

I'm pleased to inform you that your manuscript has been deemed suitable for publication in PLOS ONE. Congratulations! Your manuscript is now with our production department. 

Kind regards, 

on behalf of

Dr. Sriparna Saha 

Academic Editor

PLOS ONE